# Articulated remains of the extinct shark *Ptychodus* (Elasmobranchii, Ptychodontidae) from the Upper Cretaceous of Spain provide insights into gigantism, growth rate and life history of ptychodontid sharks

**Patrick L. Jambura**[ID]*, **Jürgen Kriwet**[ID]

Department of Palaeontology, University of Vienna, Vienna, Austria

* patrick.jambura@gmail.com

## Abstract

Due to their cartilaginous endoskeleton and the continuous tooth replacement, the chondrichthyan fossil record predominantly consists of isolated teeth, which offer diagnostic features for taxonomic identifications, but only provide very limited information of an organism's life history. In contrast, the calcified vertebral centra of elasmobranchs (sharks, skates and rays) yield important information about ecological and biological traits that can be utilized for constructing age-structured population dynamic models of extant species and palaeoecological reconstructions of such aspects in extinct groups. Here, we describe two large shark vertebrae from the Santonian (Upper Cretaceous) of Spain, which show a unique combination of characters (asterospondylic calcification pattern, with concentric lamellae and numerous parallel bands that are oriented perpendicular) that is only known from ptychodontid sharks, a distinct, extinct group of giant durophagous sharks of the Cretaceous era. Based on linear regression models for large extant sharks a total length between 430 and 707cm was estimated for the examined specimen. Our results indicate that ptychodontid sharks were large viviparous animals, with slow growth rates, matured very late and, therefore, show typical traits for K-selected species. These traits combined with a highly specialized feeding ecology might have played a crucial role for the success but also, eventually, extinction of this group.

## Introduction

More than 400 million years of evolution have shaped a diverse set of biological traits in modern elasmobranchs (sharks, skates and rays) that allowed them to occupy a variety of different niches and trophic levels. They have conquered marine and freshwater environments around the world, reaching body sizes from 0.2m (dwarf lantern shark *Etmopterus perryi*) to 20m (whale shark *Rhincodon typus*), and have developed a number of different reproductive

**Data Availability Statement:** All relevant data are within the manuscript and its Supporting Information files.

**Funding:** The authors received no specific funding for this work. Open access funding provided by University of Vienna.

**Competing interests:** The authors have declared that no competing interests exist.

strategies that can be roughly categorized in oviparity (egg laying), and viviparity (giving birth to live pups) [1,2]. Three general life history patterns can be found in sharks: 1) small body size, low longevity, small litters, small offspring, fast growth; 2) large body size, moderate to high longevity, large litters, small offspring, slow growth; 3) large body size, high longevity, small litters, large offspring, slow growth [3].

The application of life history traits has proven potentially useful in fisheries to determine if stocks are endangered and to estimate their chances to recover [4,5]. Additionally, life history traits of extinct taxa are vital for palaeoecological reconstructions and can give important insights into a species' demise, perseverance, and are crucial to augment our understanding of diversity and extinction patterns [6–8]. However, the fossil record of chondrichthyans predominantly consists of isolated teeth, which only offer limited information about the biological traits of a species. Vertebrae on the other hand yield important data on the biology and ecology of fossil elasmobranchs, but only have been described for a very limited number of extinct species [9–14].

Ptychodontid sharks seemingly were giant durophagous fish that lived in the Cretaceous period from the Aptian (~113-125mya) to the Campanian (~72-83mya) [15] and are believed to have obtained body sizes of more than ten meters [16,17]. Although fossils of this group are common in Cretaceous deposits and are known from around the globe, the taxonomic placement of this group remains ambiguous and they have been discussed to be either batomorphs [18], hybodont sharks [16], or put in the new order Ptychodontiformes within the Neoselachii (sensu Compagno [19]; Elasmobranchii sensu Maisey [20]) [21]. However, the presence of calcified vertebrae [22] and a three-layered enameloid [23] support the affinity of this group to modern sharks.

Here we describe the first articulated shark remains from the Santonian of Spain, Europe. Although no teeth were found associated with the vertebrae, taxonomic placement was possible due to a unique combination of characters that is only known for ptychodontids and allowed the exclusion of any other shark taxon known from this period. Furthermore, the vertebrae yield important information about the ontogeny, growth and body size and, therefore, provide insights into the life history of this enigmatic shark group.

## Geographic and geological setting

The material that forms the focus of this study comprises a portion of an axial skeleton consisting of five articulated and several disarticulated vertebral centra belonging to the same individual, which were collected 10 km west of Santander in northern Spain, from a section on the coast near the village of Soto de la Marina (Fig 1). Here, sediments ranging from the Cenomanian to Maastrichtian accumulating to 1200 m in thickness are well exposed. The section that yielded the shark remains starts with light greyish to whitish, massive and arenitic limestone beds forming the lower unit reaching a thickness of 24.7m [24]. The lower unit ranges from the late Campanian to early Santonian in age and is characterized by the occurrence of the holasteroid, *Cardiaster integer* ('*integer* limestones'), which is a well-known species from the calcareous platforms of the Basque-Cantabrian Region, northern and southern Pyrenees, and Alpes-Maritimes [25–27].

Within the '*integer* limestones' two layers characterized by abundant occurrences of the inoceramid, *Platyoceramus* (*Cladoceramus*) *undulatoplicatus* were identified [24,28] and designated as *undulatoplicatus* events I and II. The first appearance of *Platyceramus undulatoplicatus* marks the base of the Santonian stage [26,27,29]. The vertebrae, which were recovered from a ca. 1.2 x 0.5m large, concretionary limestone lens coming from the upper, 2nd *undulatoplicatus* Event, consequently, are of earliest Santonian age.

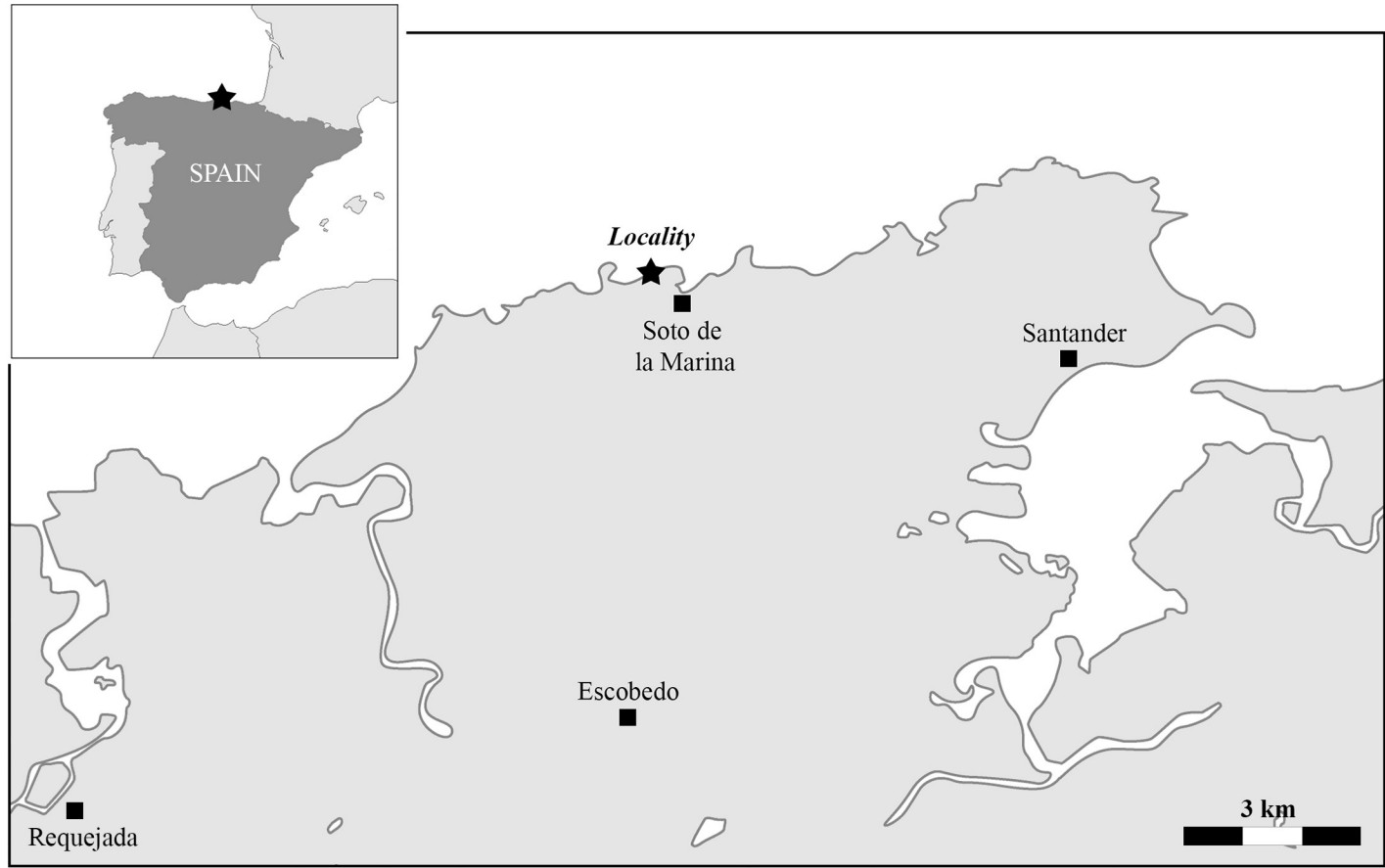

**Fig 1. Fossil locality near Santander, northern Spain where EMRG-Chond-SK-1 was recovered (indicated by a black star).**

## Material and methods

### Documentation and preparation

The articulated vertebral section was photographically documented in the field but not collected (Fig 2). Two disarticulated vertebral centra of varying degree of preservation were recovered and small sediment samples adjacent to the incomplete vertebral column were taken for screen-washing. The material was collected by a diploma student (Kurt Oppermann, Berlin) within a larger research project of the late Prof. G. Ernst (†25.04.2002) at the Free University of Berlin in 1996 when no additional permits were required. The centra are publicly accessible and housed in the fossil vertebrate collection of the Department of Palaeontology (University of Vienna) under the collection number EMRG-Chond-SK-1. Of one vertebral centrum, a dorso-ventrally directed thin section was prepared. The sediment samples and carbonate matrix of the other specimens were removed with 10% acetic acid and the residues sieved with a 0.25 mm mesh and sorted under a binocular. The recovered material comprises some dermal scales of the elasmobranch placoid type, which are deposited in the fossil collection of the Institute of Palaeontology, Free University Berlin without numbers. Some of these placoid scales were studied with a Jeol high-vacuum scanning electron microscope at the Institute of Geological Sciences of the Free University Berlin, Germany (S1 Fig).

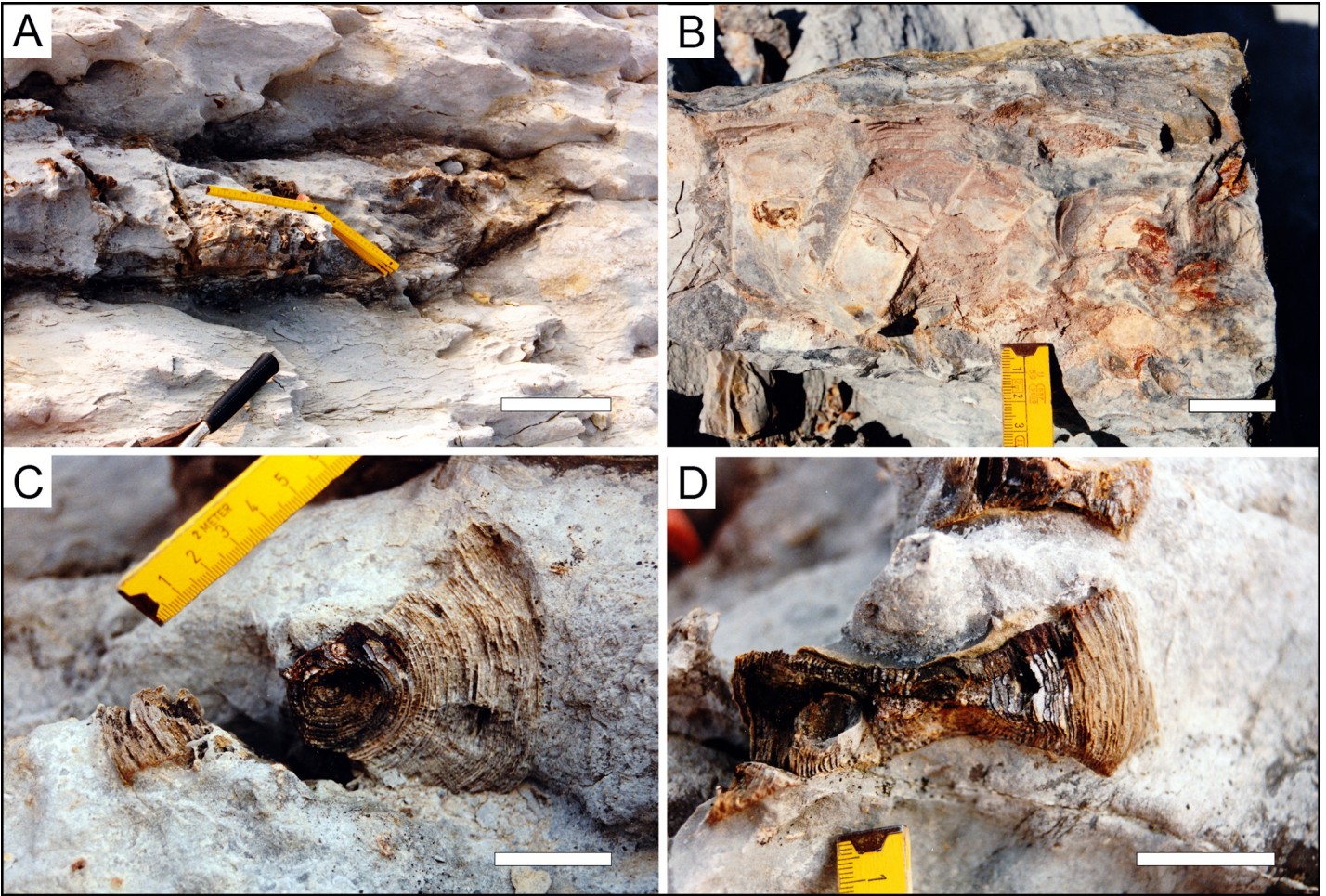

**Fig 2.** Additional articulated (A, B) and disarticulated shark vertebrae (C,D) found in situ associated with EMRG-Chond-SK-1. Picture courtesy of K. Oppermann, 1997. Scale bars equal 20cm (plate A) and 3cm (plates B, C, D).

### Size estimation

Previous studies showed a linear relationship between total length (TL) and vertebral centrum diameter (CD) in several shark species [30–34] that can be expressed with a regression equation [10,35]. Due to the fragmentary nature of the material, we measured the radius and multiplied it by two to obtain the diameter of the centrum. The larger vertebra EMRG-Chond-1b had a sedimentary infill between increment rings 25 and 26. Taking this into account, the radius was measured to the 25th increment ring and the distance between the 26th and outermost increment ring (31) was added. The distance between band pair 25 and 26 was estimated as the average value of the band intervals of the five preceding and five succeeding increment rings. As it was impossible to determine the position of the two isolated vertebrae, we conservatively regarded the vertebra with the higher centrum diameter as the largest vertebra in the entire individual. This approach ensured a minimum estimate for the total length and reduced the risk of overestimating the body size.

Most pelagic sharks have a consistent body shape [10,36], therefore, it was regarded as reasonable in previous studies to assume that the relationship between vertebral size and total body length is consistent between species with similar ecologies. However, it has been reported

that this relationship can vary between species [9,10,12]. To overcome this issue, we conducted two independent approaches to estimate the total length of the examined specimen:

(1) We extrapolated the total length of EMRG-Chond-SK-1b by comparing it with the estimated total length (TL) and centrum diameter (CD) of †*Ptychodus occidentalis* [16].

$$TLx(cm) = TL_{\dagger Ptychodus\ occidentalis} \times \frac{CD_{EMRG-Chond-SK-1b}}{CD_{\dagger Ptychodus\ occidentalis}}$$

$$TLx(cm) = X \times \frac{CD_{EMRG-Chond-SK-1b}(mm)}{11.5mm} \quad X \in \{150cm;\ 200cm\}$$

(2) We used published regression equations for large extant shark species with comparably sized vertebrae to estimate the relationship between centrum diameter (CD; mm) and total length (TL; cm). The following species with known regression equations were used as templates: (1) the great white shark *Carcharodon carcharias* [35]; (2) tiger shark *Galeocerdo cuvier* [37]; (3) whale shark *Rhincodon typus* [34].

$$TL_{Carcharodon\ carcharias} = 22 + 5.8 \times CD$$

$$TL_{Galeocerdo\ cuvier} = 35.293 + 14.314 \times CD$$

$$TL_{Rhincodon\ typus} = 36.695 + 9.531 \times CD$$

## Results and discussion

### Systematic palaeontology

Class CHONDRICHTHYES Huxley, 1880 [38]
Subclass ELASMOBRANCHII Bonaparte, 1838 [39]
Order *incertae sedis*
Family PTYCHODONTIDAE Jaekel, 1898 [18]
Genus *PTYCHODUS* Agassiz, 1835 [40]

*Diagnosis (emended)*. For dental characters of this genus see Hamm [41]. Vertebral centra are well calcified, amphicoelous and circular, not dorso-ventrally compressed. Vertebrae display a calcification pattern of the asterospondylic type with four uncalcified areas radiating diagonally from the center to the bases of the neural and haemal arches. Numerous concentric lamellae extend outwards from the center. Underneath the smooth articular surfaces, parallel lamellae are oriented 360° around the center of the vertebrae and run perpendicular to the concentric lamellae.

*Material*. EMRG-Chond-SK-1; two vertebral centra.

*Locality*. Soto de la Marina, west of Santander, Cantabria, N Spain.

*Age and horizon*. Early Santonian, 'integer limestone', *undulatoplicatus* Events II.

*Description*. Both centra were found articulated with several other vertebrae, which were not recovered and left in the field (Fig 2). Vertebra centra show a marked concavity on the articular surfaces (amphicoelous) and appear nearly circular in median transverse view. The dimensions are approximately 55mm (EMRG-Chond-SK-1a) and 70mm (EMRG-Chond-SK-1b) in diameter (dorsoventrally) and the vertebral anterior-posterior length is 23mm (EMRG-Chond-SK-1a). Due to the fragmentary condition of the material, no dimensions could be measured for the mediolateral diameter.

Shark centra form a double-cone calcification with densely calcified anterior and posterior conical ends, collectively referred to as corpora calcarea. Between the corpora calcarea is the

intermedialia, which is softer than the corpus calcareum. The articular facet of the corpus cal-careum is weathered and expose the inner layer of the vertebrae, showing concentric calcare-ous rings extending outwards from the center of the vertebrae. Numerous parallel lamellae are oriented perpendicular to these concentric lamellae (Fig 3A). EMRG-Chond-SK-1b was sec-tioned transversely and exhibits an asterospondylic calcification pattern (Fig 3B): secondary calcification leaves four uncalcified areas (i.e. two basidorsal and two ventral cartilage wedges), which radiate diagonally from the center to the base of the neural and haemal arches. Concen-tric lamellae are restricted to the area between the wedges. A total of 31 growth increments can be identified in the vertebra section. Post-mortem sedimentary infilling can be observed between ring 25 and 26 which inflated the distance between those two rings (Fig 3B, S2 Fig). The dorsoventral and mediolateral radii of the first increment ring (birth ring) are 3.6 and 3.2mm, respectively.

## Taxonomic remarks

In contrast to teeth, vertebral centra are thought to bear only little taxonomic information for extinct elasmobranch fishes as comparative analyses are hard to perform due to the lack of articulated material with associated teeth. Hasse [42] recognized three different calcification patterns of vertebrae in cross-section (cyclospondyl, tectospondyl, and asterospondyl), which later was revised by Ridewood [43], who stated that these three categories were not sufficient to describe the plethora of different calcification patterns that can be found in sharks, skates and rays. The vertebral centrum EMRG-Chond-SK-1b displays the asterospondylic type (sensu Hasse [42]) with four uncalcified areas radiating from the center to the bases of the neu-ral and haemal arches, which is typical for galeomorph sharks [44].

   Another character that is apparent in the cross section is the presence of concentric lamellae that are extending outwards from the center. The combination of these two features (asteros-pondyly with uncalcified wedges and concentric lamellae) are only known from the basking shark *Cetorhinus maximus* [43,45], the whale shark *Rhincodon typus* [31], †*Ptychodus* [46,47], and †*Squalicorax* [47,48]. Both †*Ptychodus* and †*Squalicorax* are known from Cretaceous deposits in Europe, N- and S-America, Africa and Asia [15]. Our specimen shows numerous parallel bands that are oriented 360˚ around the center of the vertebrae. These parallel lamellae are oriented perpendicular to the concentric lamellae, a trait only known from ptychodontid sharks and is regarded as a diagnostic feature for this group, which is absent in other sharks, including *C. maximus*, *R. typus*, and †*Squalicorax* [21,49]. This assumption is also supported by previous reports of ptychodontid shark vertebrae, which display this feature [22,46,50]. Rozefelds [51] reported large vertebral centra from the lower Cretaceous Toolebuc Formation of Australia, which resemble our specimens. He found associated placoid scales but no oral teeth and assigned the species tentatively to the anacoracid genus †*Pseudocorax*. This seems very unlikely because †*Pseudocorax* was a rather small shark with tooth crown heights of 1.5cm [15], which is comparable to the teeth of a two meter long †*Squalicorax falcatus* [9]. Fur-thermore, like in our specimen, parallel lamellae are visible, which is not known from anacora-cid sharks. The combination of the above mentioned characters (parallel lamellae, concentric lamellae, asterospondyl centra), the size of the vertebral centra and the stratigraphic age (Cre-taceous) of these species, leads us to the assumption that both specimens, EMRG-Chond-SK-1 and Rozefelds' QMF18264, are ptychodontid sharks. Other big sharks from the Late Creta-ceous are known from the order Lamniformes (e.g., †*Cretalamna*, †*Cretodus*, †*Cretoxyrhina*), but can easily be ruled out because they are known to have a different mineralization pattern of the vertebral centra (i.e., the vertebral centrum is strengthened by multi-branched, densely packed lamellae), have radial lamellae on the dorsoventral axis in lateral view (which our

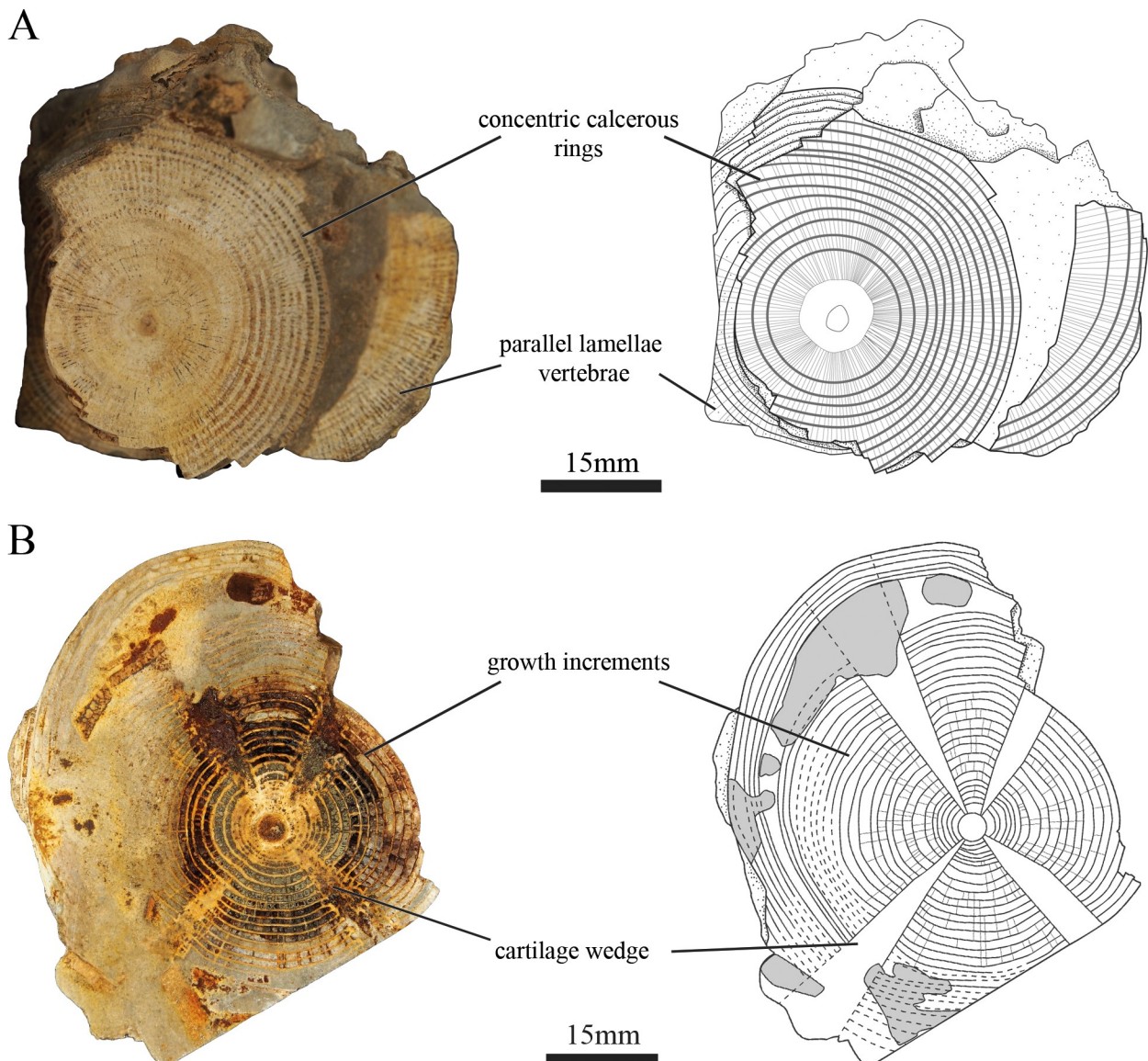

**Fig 3. Calcification pattern of the vertebral centra of EMRG-Chond-SK-1.** (A) close up view and illustration of the vertebral centra EMRG-Chond-SK-1a. The articular surface is weathered and exposes the inner layer of the vertebrae, showing concentric calcareous rings and parallel bands that run perpendicular to them; (B) vertebral section in transverse plane and illustration of EMRG-Chond-SK-1b show an asterospondylic calcification pattern with four cartilage wedges radiating diagonally from the center to the base of the neural and haemal arches.

specimen does not have), and lack parallel lamellae [12,13,48,52–55]. The placoid scales from the sediment samples associated with the Spanish specimen represent three different morphotypes (see S1 Fig). Placoid scale type 3 of the Spanish specimen resembles those figured by Shimada et al. [16,17] and the 'curved crown scale' type of Rozefelds [51] to some extent. The other two scale types in the specimen described here have not been figured before. These morphological differences might be related to different positions of the scales on the body. As no teeth were found associated with this specimen, an exact taxonomic identification on species level remains impossible, which leaves us to refer to it as †*Ptychodus* sp.

## Age, growth and inferred life history traits

**Age estimation.** Whole vertebral centra, as well as transverse and sagittally sectioned centra have been commonly used for aging elasmobranchs by counting alternating opaque and translucent band pairs (also referred to as band pairs, annuli, rings, or vertebral growth increments) that are deposited in the vertebral centra as they grow [10,21,56–58]. A number of studies suggested that these band pairs are deposited annually in several elasmobranch species e.g., in the shortfin mako shark *Isurus oxyrinchus* [32], scalloped hammerhead shark *Sphyrna lewini* [59], dusky shark *Carcharhinus obscurus* [60], leopard shark *Triakis semifasciata* [61], smalltooth sawfish *Pristis pectinata* [62], etc. [63–71]. However, this is questioned by recent studies [72–74] and especially in old individuals of long living species the age determination seems to be highly underestimated as the growth rate decreases with age resulting in a band width decline at the centrum edges that can become unresolvable in older individuals [56,72,75–79]. The section of EMRG-Chond-SK-1b revealed 31 band pairs (birth mark + 30 band pairs), suggesting, under the assumption of an annual growth band deposition, that EMRG-Chond-SK-1 was around 30 years old. These band pairs were well distinguishable, indicating that EMRG-Chond-SK-1 had not reached the maximum length when it died, because no compressed arrangement of band pairs is preserved at the edges.

**Body size estimation.** Based on the previously published centrum diameter and estimated total length of †*Ptychodus occidentalis* [16] we calculated an estimated total length of 887-1183cm for EMRG-Chond-SK-1. However, this estimation should be taken with caution, as the TL-CD relationship of †*P. occidentalis* is based on a single vertebral centrum which not necessarily represents the largest vertebra in this specimen and, therefore, can result in overestimated size approximations. Therefore, we recommend taking the previously estimated TL of 13m for †*P. rugosus* [16], which was also based on this TL-CD relationship, with much caution. Shimada et al. [16] also compared the anterior-posterior length of the teeth of †*P. occidentalis* and †*P. rugosus* and concluded that †*P. rugosus* might have reached a body size of even 14.4m. However, it is important to note here that the total length for †*P. occidentalis*, on which both calculations are based on, is unknown and was estimated based on the length of the lower jaw. Therefore, an erroneous size estimation for †*P. occidentalis* would also bias all subsequent calculations.

Further indication of overestimated body sizes for †*Ptychodus* is given by our calculations of the total length of EMRG-Chond-SK-1 based on regression equations for large extant shark species. In contrast to the above-mentioned extrapolation, this approach has the advantage, by assuming EMRG-Chond-SK-1b to be the largest vertebra, to offer minimum size estimations and, therefore, reducing the risk of overestimating the size. Using the vertebral diameter of 70 mm and the regression equations for the great white shark *Carcharodon carcharias* [35], tiger shark *Galeocerdo cuvier* [37] and whale shark *Rhincodon typus* [34], the minimum total length of EMRG-Chond-SK-1 is calculated to be 430cm, 539cm, and 707cm respectively (Fig 4). The use of regression equations for three different species (from three different orders) has shown significant variations in estimated body sizes and, thus, indicates that previous assumptions of the more or less consistent body forms in pelagic sharks resulting in similar size estimations [36] were oversimplified. In fact, a variety of different factors (e.g., phylogenetic affiliations, lifestyle, trophic level, etc.) might contribute to the relationship between vertebral diameter and total body length and, therefore, affect size estimations. However, it seems reasonable to assume that the total length of EMRG-Chond-SK-1 lies within the estimated range of 430-707cm as the three template species of the regression equations cover a wide range of different ecologies (microphagous and macrophagous sharks) and are not closely related to each other (diversification of these three groups occurred in the Early and Middle Jurassic, respectively

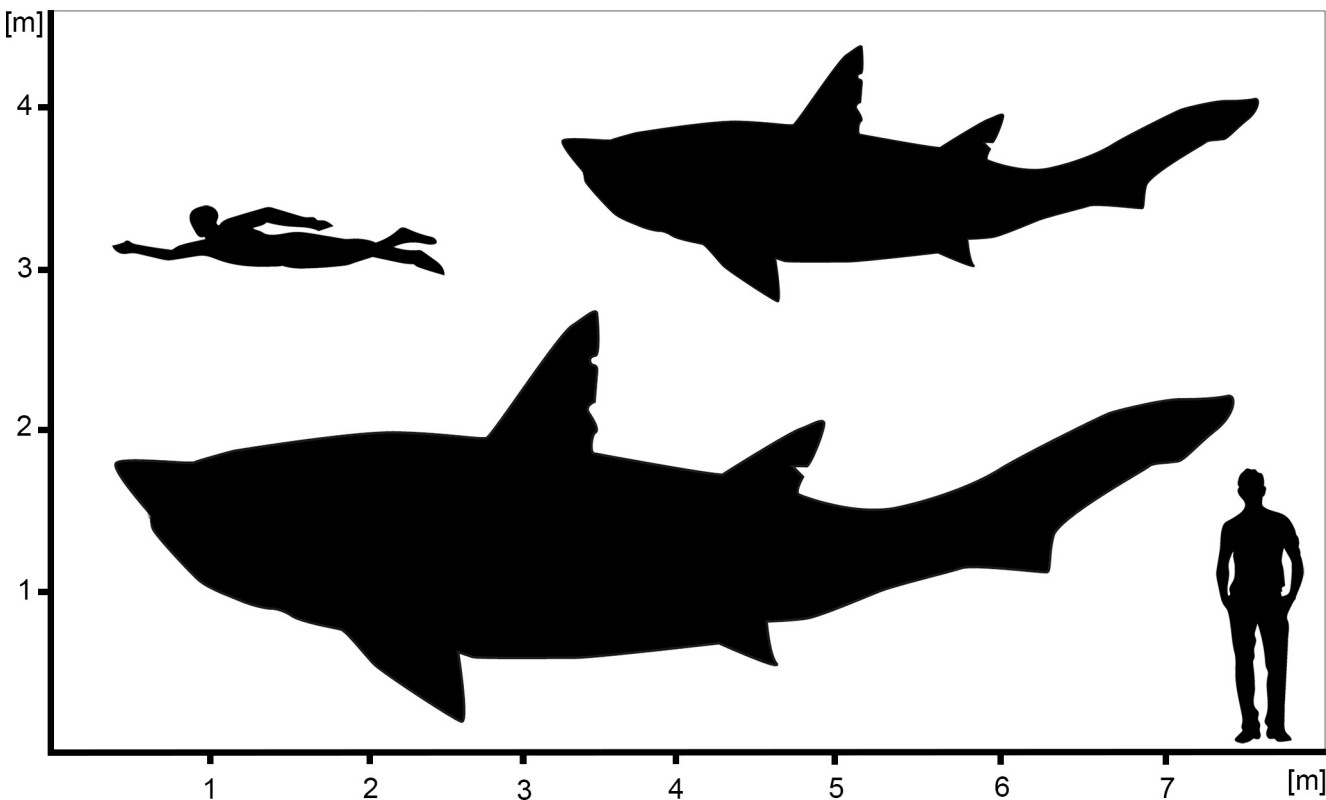

**Fig 4. Estimated total length for the examined ptychodontid shark from Spain.** Hypothetical outlines of †*Ptychodus* sp. showing the minimum and maximum size estimations for the specimen EMRG-Chond-SK-1.

[8]). Given that our specimen most likely has not yet reached maturity and therefore represents a subadult, previous size estimations of around 10m [17] seem possible for this group. Although more accurate maximum size estimations need to wait until a complete specimen can be analyzed, our study agrees with previous work that †*Ptychodus* was one of the largest durophagous vertebrates ever to have lived.

**Growth rate.** The intervals between adjacent band pairs remain more or less stable from the innermost band pair to the outermost band pair, although some variability does exist (Table 1). This differs from previous studies [10–13], which reported the intervals to decrease from the innermost to the outermost band pairs and indicates that the growth rate of EMRG-Chond-SK1 has not decreased until its death (Fig 5A). Attempts to fit the data to a von Bertalanffy growth model resulted in erroneous maximum size estimations ($TL_{max}$>2000cm). Plotting the centrum radius CR of each growth ring against the growth ring count ("age") resulted in a graph that followed a linear model which, because of the linearity between CR/CD and TL, follows the same trend if TL is plotted against growth ring count (or "age") (Fig 5B). Most growth models for fish are nonlinear [80] and shark growth models are usually best described by an asymptotic growth model (e.g., von Bertalanffy growth model) [58]. To date, all examined sharks and rays show asymptotic growth during ontogeny and it therefore seems justified to assume that †*Ptychodus* did so as well. Our data suggest that EMRG-Chond-SK1 has not reached the plateau of the asymptote of the growth curve yet and, consequently, not its maximum length. Furthermore, clear inflections of the growth curve, that indicate a decreased growth rate (e.g., after reaching maturity when energy from somatic growth is diverted to gonadal growth), are absent in the data set from specimen EMRG-Chond-SK1 and, thus,

**Table 1. Measurements and derived estimations taken from vertebra EMRG-Chond-SK-1b.**

| BN | CR (mm) | BI (mm) | pCD (%) | $TL_{CC}$ (cm) | $TL_{GC}$ (cm) | $TL_{RT}$ (cm) |
|---|---|---|---|---|---|---|
| 1 | 3.7 | - | 10.51% | 64.92 | 88.25 | 107.23 |
| 2 | 4.4 | 0.7 | 12.50% | 73.04 | 98.28 | 120.57 |
| 3 | 5.3 | 0.9 | 15.06% | 83.48 | 111.16 | 137.72 |
| 4 | 6.1 | 0.8 | 17.33% | 92.76 | 122.61 | 152.97 |
| 5 | 6.9 | 0.8 | 19.60% | 102.04 | 134.06 | 168.22 |
| 6 | 8.1 | 1.2 | 23.01% | 115.96 | 151.24 | 191.10 |
| 7 | 9.5 | 1.4 | 26.99% | 132.20 | 171.28 | 217.79 |
| 8 | 10.5 | 1 | 29.83% | 143.80 | 185.59 | 236.85 |
| 9 | 12 | 1.5 | 34.09% | 161.20 | 207.06 | 265.44 |
| 10 | 12.9 | 0.9 | 36.65% | 171.64 | 219.94 | 282.60 |
| 11 | 14.2 | 1.3 | 40.34% | 186.72 | 238.55 | 307.38 |
| 12 | 15.6 | 1.4 | 44.32% | 202.96 | 258.59 | 334.06 |
| 13 | 16.8 | 1.2 | 47.73% | 216.88 | 275.77 | 356.94 |
| 14 | 18 | 1.2 | 51.14% | 230.8 | 292.95 | 379.81 |
| 15 | 19.3 | 1.3 | 54.83% | 245.88 | 311.55 | 404.59 |
| 16 | 20.4 | 1.1 | 57.95% | 258.64 | 327.30 | 425.56 |
| 17 | 21.4 | 1 | 60.80% | 270.24 | 341.61 | 444.62 |
| 18 | 22.5 | 1.1 | 63.92% | 283.00 | 357.36 | 465.59 |
| 19 | 24.1 | 1.6 | 68.47% | 301.56 | 380.26 | 496.09 |
| 20 | 25.3 | 1.2 | 71.88% | 315.48 | 397.44 | 518.97 |
| 21 | 26.4 | 1.1 | 75.00% | 328.24 | 413.18 | 539.94 |
| 22 | 27.2 | 0.8 | 77.27% | 337.52 | 424.63 | 555.19 |
| 23 | 28.1 | 0.9 | 79.83% | 347.96 | 437.52 | 572.34 |
| 24 | 29 | 0.9 | 82.39% | 358.40 | 450.40 | 589.50 |
| 25 | 29.9 | 0.9 | 84.94% | 368.84 | 463.28 | 606.65 |
| 26 | N/A | N/A | N/A | N/A | N/A | N/A |
| 27 | N/A | 0.8 | N/A | N/A | N/A | N/A |
| 28 | N/A | 0.7 | N/A | N/A | N/A | N/A |
| 29 | N/A | 1.1 | N/A | N/A | N/A | N/A |
| 30 | N/A | 1.1 | N/A | N/A | N/A | N/A |
| 31 | *35.2 | 0.8 | 100.00% | 430.32 | 539.15 | 707.68 |

**Abbreviations: BN**, band number; **CR**, centrum radius; **BI**, band interval; **pCD**, percent centrum diameter from the center of the vertebra $TL_{CC}$, total length estimation based on the regression equation of *Carcharodon carcharias* [35]; $TL_{GC}$, total length estimation based on the regression equation of *Galeocerdo cuvier* [37]; $TL_{RT}$, total length estimation based on the regression equation of *Rhincodon typus* [34].

*reconstructed value; see "Materials and methods" for more details on its computation.

suggests that this individual has not reached maturity at band pair 25 with an estimated body size between 369 and 607cm. These estimations are reasonable when compared to modern giant sharks ("gigantism" sensu Pimiento *et al*. [81] refers to sharks with body sizes exceeding six meters), which show similar sizes at maturity, e.g., the great white shark, *Carcharodon carcharias*, at 350-500cm (TL about 600cm), basking shark, *Cetorhinus maximus*, at 400-800cm (TL more than 1000cm), whale shark, *Rhincodon typus*, at 600-800cm (TL 1700-2100cm) [2]. When compared to big macropredatory sharks (i.e., great white shark *Carcharodon carcharias* and *Cretoxyrhina mantelli*), it is apparent that the slope of the growth curve of †*Ptychodus* is less steep and more similar to the microphagous basking shark *Cetorhinus maximus* (Fig 5C). Therefore, the growth rate of †*Ptychodus* is assumed to be lower than those of

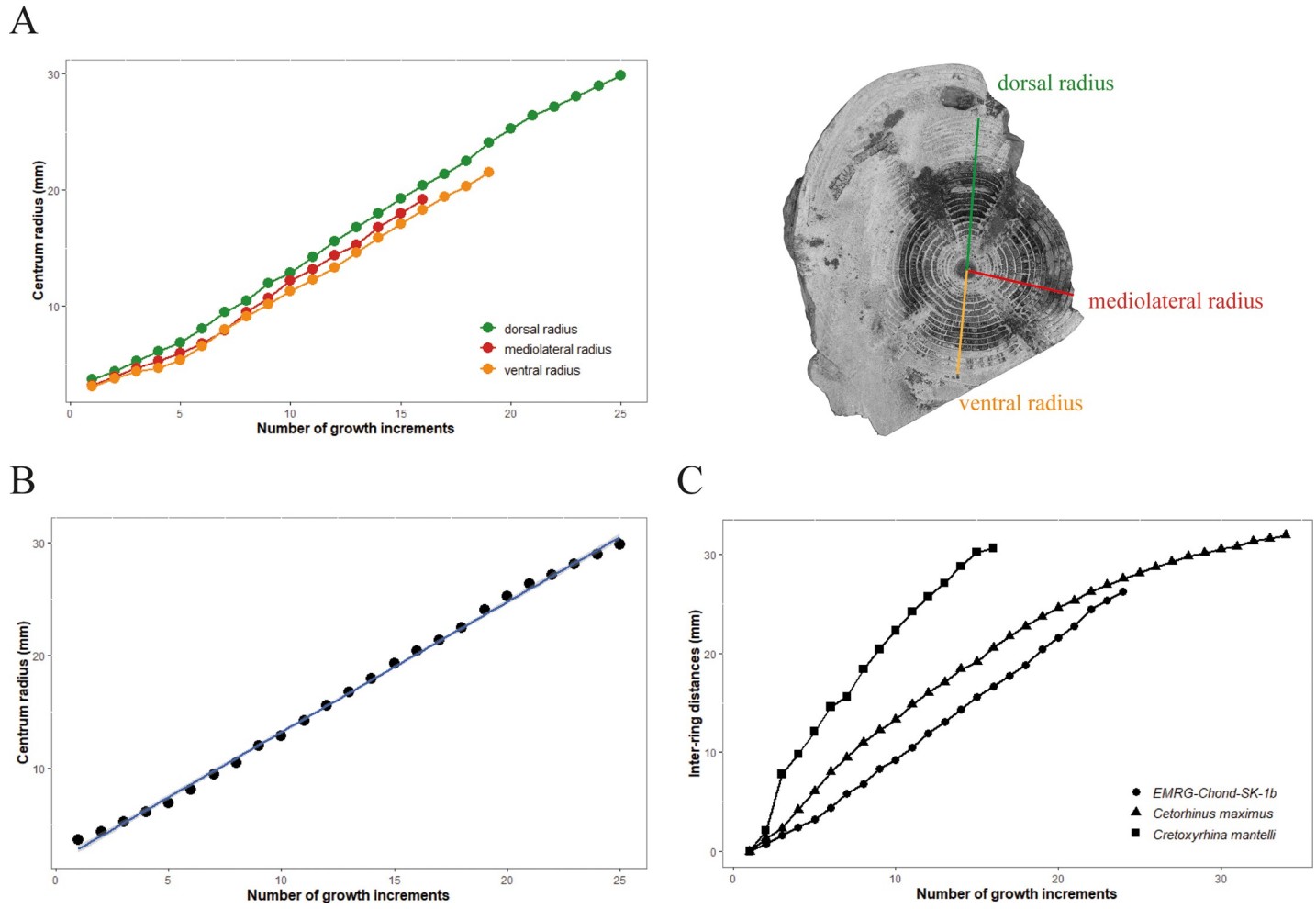

**Fig 5. Growth ring profile of EMRG-Chond-SK-1b.** (A) centrum size profiles of EMRG-Chond-SK-1b follow the same trend for radii in dorsal, ventral and mediolateral direction. (B) data points of the centrum size profile closely follow a linear function. (C) growth ring distances of EMRG-Chond-SK-1b compared to †*Cretoxyrhina mantelli* [10] and *Cetorhinus maximus* [82].

apex predatory sharks. Under the assumption of an annual growth band deposition, †*Ptychodus* matured very late (after more than 25 years) and showed great longevity, similar to the giant filter-feeding sharks that live today [34,82].

**Inferred life history.** Vertebral centra of sharks have proven useful for estimating body sizes [13,36], growth rates [10–13,32,33,69] and, therefore, allow the reconstruction of biological and ecological traits, as various of these aspects of an organism are correlated with body size [83–86]. Although we could not determine the identity of EMRG-Chond-SK-1 on species level, it gives us valuable insights into the ontogeny and ecology of ptychodontid sharks. Our analysis on the vertebral growth indicates that †*Ptychodus* was slow growing, late maturing and seemingly long-living, all of which are key traits for a K-selected species. Based on the radius of the birth ring (3.7mm) we estimated a total length of 65-107cm for our specimen at the time of birth. Offspring with similar sizes are known from a variety of large viviparous sharks, e.g. great hammerhead shark *Sphyrna mokarran* (50-70cm), tiger shark *Galeocerdo cuvier* (51-76cm), great white shark *Carcharodon carcharias* (110-160cm), basking shark *Cetorhinus maximus* (150-170cm), and whale shark *Rhincodon typus* (55-64cm) [2]. Similar sizes at birth have not been reported for oviparous sharks (usually not exceeding 15-25cm), which

leads us to the assumption that †*Ptychodus* also was a viviparous shark that put a lot of resources into the development of large offspring.

To date, we can only speculate about the reasons for the extinction of this group. However, K-selected species are characterized by specific adaptations (slow growing, late maturity, large body, small size of litter) that make such sharks more prone to environmental changes and have been correlated with increased extinction risk compared to oviparous (r-selected) sharks [87]. Our results strongly suggest that extinct ptychodontid sharks had K-selected traits, which in combination with a highly specialized trophic niche (durophagy) might have been major intrinsic contributors to the demise of this group.

## Supporting information

**S1 Fig. Placoid scales found associated with vertebrae of EMRG-Chond-SK-1.** (A) type 1, "six-keeled scales"; (B) type 2, "three-keeled scales"; (C) type 3, "knob-like scales". Scale bar equals 100μm.
(TIF)

**S2 Fig. Transverse section of EMRG-Chond_SK-1b, showing the number of growth increments.**
(TIF)

## Acknowledgments

We are deeply indebted to Kurt Oppermann (Berlin, Germany) for the possibility to study the vertebral centra and placoid scales of this interesting extinct shark, and for providing geographic and geological information. We are grateful to Manuel Amadori and Sebastian Stumpf (both from the University of Vienna, Austria) for discussions and to Kenshu Shimada for providing pictures of an articulated skeleton of *Squalicorax falcatus* which comprises 86 vertebrae. We also want to thank Jacopo Amalfitano and Giuseppe Marramà, for their constructive comments on an earlier version of the manuscript and Giorgio Carnevale for editorial comments that helped improving a previous version of this manuscript. Open access funding provided by University of Vienna.

## Author Contributions

**Conceptualization:** Patrick L. Jambura.

**Data curation:** Patrick L. Jambura, Jürgen Kriwet.

**Formal analysis:** Patrick L. Jambura.

**Funding acquisition:** Patrick L. Jambura.

**Investigation:** Patrick L. Jambura.

**Methodology:** Patrick L. Jambura.

**Resources:** Jürgen Kriwet.

**Software:** Patrick L. Jambura.

**Supervision:** Jürgen Kriwet.

**Validation:** Patrick L. Jambura, Jürgen Kriwet.

**Visualization:** Patrick L. Jambura.

**Writing – original draft:** Patrick L. Jambura, Jürgen Kriwet.

**Writing – review & editing:** Patrick L. Jambura, Jürgen Kriwet.

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
