## [Decision Letter · Decision Letter 0]

18 Mar 2020

PONE-D-20-05473

First articulated remains of the extinct shark, Ptychodus (Elasmobranchii, Ptychodontidae) from the Upper Cretaceous of Spain provide insights into gigantism, growth rate and life history of ptychodontid sharks

PLOS ONE

Dear Dr. Jambura,

Thank you for submitting your manuscript to PLOS ONE. After careful consideration, we feel that it has merit but does not fully meet PLOS ONE’s publication criteria as it currently stands. Therefore, we invite you to submit a revised version of the manuscript that addresses the points raised during the review process.

Both the reviewers find the paper interesting and acceptable pending minor changes. In particular, as suggested by the second reviewer, additional data from extant and fossil galeomorphs for comparative purposes would be desirable.

We would appreciate receiving your revised manuscript by April 16 2020. To enhance the reproducibility of your results, we recommend that if applicable you deposit your laboratory protocols in protocols.io, where a protocol can be assigned its own identifier (DOI) such that it can be cited independently in the future. For instructions see: http://journals.plos.org/plosone/s/submission-guidelines#loc-laboratory-protocols

We look forward to receiving your revised manuscript.

Kind regards,

Giorgio Carnevale, Ph.D

Academic Editor

PLOS ONE

Reviewers' comments:

Reviewer's Responses to Questions

**Comments to the Author**

1. Is the manuscript technically sound, and do the data support the conclusions?

Reviewer #1: Yes

Reviewer #2: Yes

2. Has the statistical analysis been performed appropriately and rigorously? 

Reviewer #1: Yes

Reviewer #2: N/A

3. Have the authors made all data underlying the findings in their manuscript fully available?

Reviewer #1: Yes

Reviewer #2: Yes

4. Is the manuscript presented in an intelligible fashion and written in standard English?

Reviewer #1: Yes

Reviewer #2: Yes

5. Review Comments to the Author

Reviewer #1: Dear Editor and Authors,

The study reports some interesting findings about the enigmatic genus Ptychodus from the Cretaceous of Spain and has a clear paleobiological approach. Despite the fossil material is fragmentary, it displays peculiar and useful characters. These characters have never been remarked in other papers and allow the authors to make some very interesting paleobiological inferences. Data are presented in an appropriate fashion and statistics are adequate, even though the Von Bertalanffy growth model was not applicable (but the authors clearly state the reason of its unsuitability). The inferences are well-supported and the language is fluent. All in all, this study is of international interest and provides novel and significant information.

There are only some minor points I would like to raise:

- The title deals with first articulated specimens from Spain, but there is no mention in the manuscript to other findings of the genus Ptychodus from Spain. It would be useful to provide some other references to these findings and compare with the specimens reported herein.

- The diagnostic vertebral characters reported here should be included in the diagnosis of the genus (see Hamm, 2020 and please emend).

- Please reconsider the structure of the manuscript and the issues with the images and other minor indications addressed in the annotated version of the manuscript attached.

I would be pleased to answer any question from the authors and be open to any discussion.

Best regards,

Jacopo Amalfitano, PhD

Reviewer #2: The manuscript of Jambura & Kriwet is an interesting and important contribution to the knowledge of the paleobiology of the enigmatic extinct shark Ptychodus. The manuscript is of high-impact, well written, and provides sufficient data to suggest new hypotheses about the palaeobiology of Ptychodus. However, the manuscript still presents some issues that the authors should address before the publication. A further proofcheck is also needed to avoid small typos and errors. This is why I consider the manuscript worth of publication after minor revision. In particular:

- In the title and several times in the text (e.g. line 70) the authors emphasize that these are the first articulated Ptychodus remains. However, there are other specimens of Ptychodus described in the literature that can be considered as articulated, and these include nearly complete tooth sets (e.g. Amadori et al. 2019), and even articulated vertebrae (Hamm 2010, fig. 6B). This is why authors should remove ‘first’ from the title and subsequent statements in the manuscript.

- The authors used regression equations for extant galeomorph sharks to estimate the relationships between centrum diameter and total length of Ptychodus. However, the authors should clearly state, in Material and Methods, the reason why they use galeomorph sharks as comparative taxa, and not squalomorphs, or other extinct groups. As far as we know the affinities of Ptychodus are far from being clear. Is the presence of asterospondylic vertebrae enough to detect its close relationship with galeomorph sharks? Are there other hypotheses in the literature to be considered? If so why are these hypotheses discarded?

- Moreover, the authors use Carcharodon carcharias, Galeocerdo cuvier, Rhincodon typus as the only representatives of galeomorph sharks for comparisons. However, there are other extinct and living galeomorphs for which the relationship between centrum diameter and total body length are known, as reported by the authors themselves in the literature, that should be considered. For example: the extant Prionace glauca, Isurus oxyrhincus, Carcharhinus limbatus, and the extinct Carcharodon megalodon, and Cretoxyrhina mantelli (see Stevens 1975; Killam & Parsons 1989; Gottfried et al. 1996; Ribot-Carballal et al. 2005; Shimada 2008). There are probably even more taxa. In my opinion, the authors should consider for comparison the higher number of galeomorph taxa as possible and discuss, in any case, the results obtained. Otherwise, they should clearly state in Material and Methods the reason why they limited their comparisons to these three particular taxa.

- Although it is not a common rule, several journals suggest to avoid the Saxon genitive in scientific papers being this mostly used in colloquial and informal sentences. I would suggest to avoid the Saxon genitive also here (see lines 27, 39, 57, 282, 312, 351).

- Line 46. Replace “reproduction strategies” with “reproductive strategies”.

- Line 47. Some lamniform and carcharhiniform sharks and some rays (e.g. electric rays and stingrays) are actually ovoviviparous (aplacental viviparity). In my opinion this should be considered a different, third type of reproductive mode since it is quite different from the pure viviparity in that there is no placental connection and the unborn young are usually nourished by egg yolk.

- Line 50. Do you mean ‘slow growth’ instead of ‘small growth’?

- Line 104 to 107. This sentence is unclear and/or seems incomplete. Please re-write it.

- Line 134. I would write ‘conservative body shape’ instead of ‘consistent body form’, being ‘shape’ only related to the ‘outline’ but not to size.

- Line 141. It is unclear to me how the authors extrapolate the equation (1), or if this is based on a published paper. If the goal is using the following proportion:

TLEmrg-Chond-SK-1b : TLP.occidentalis = CDEmrg-Chond-SK-1b : CDP.occidentalis

you should consider that the product of the means (TLP.occidentalis x CDEmrg-Chond-SK-1b ) equals the product of the extremes (TLEmrg-Chond-SK-1b x CDP.occidentalis ) and then you must employ this equation:

TLemrg-Chond-SK-1b = (TLp.occidentalis x CDemrg-Chond-SK-1b) / CDp.occidentalis

Please also show somewhere which are the values from the published literature that you use for the equations. That means, just replace abbreviations (TL, CD) with numbers.

- Line 222-223. Should it maybe be ‘…and lack parallel lamellae ON vertebrae’ ?

- Line 255-256. ‘single vertebral centrum’ and ‘the largest vertebra’

- Line 258. In Shimada et al. (2009) it seems that the body size estimate for P. rugosus is not calculated based on vertebral centrum diameter, but rather on the antero-posterior tooth crown length. Please check.

- Line 290. Please spell the first time what CR is.

- In the whole manuscript, please be consistent in using meters or centimetres to indicate the total body length; and millimetres or centimetres for the radius/diameter of the centra.

- Line 346-347. A bit unclear. What about ‘However, K-selected species are characterized by specific adaptations… etc’

- Line 347/348. Replace ‘changing environments’ with ‘environmental changes’.

- Line 349. Be careful. You DID NOT demonstrate unambiguously that ptychodontids had K-selected traits (although you can assume/hypothesize it) because your hypothesis has been inferred based on indirect evidences and/or comparisons with living representatives, not with statistical demonstration or direct observations. Maybe better the sentence as 'We suggest/ can infer/ hypothesize that..."

- In figure 3, the names of the anatomical features should be in ‘lower case’.

- Line 498. The first author surname is ‘Larocca Conte’.

Suggested literature:

Hamm SA. 2010. The Late Cretaceous shark, Ptychodus rugosus , (Ptychodontidae) in the Western Interior Sea. Transactions of the Kansas Academy of Science, 113: 44-55.

Killam KA., Parsons, GR. 1989. Age and growth of the blacktip shark, Carcharhinus limbatus, near Tampa Bay, Florida. Fishery Bulletin. U.S. 87: 845-857.

6. PLOS authors have the option to publish the peer review history of their article (what does this mean?). If published, this will include your full peer review and any attached files.

Reviewer #1: Yes: Jacopo Amalfitano

Reviewer #2: Yes: Giuseppe Marrama'

---

## [Author Response · Author response to Decision Letter 0]

25 Mar 2020

Dear Editor and reviewers.

First, let me thank you, also in the name of my co-author, Jürgen Kriwet, for all the work and effort that you put into this manuscript to make it ready for publication. We went through all your comments and, except for a few issues, agreed with your suggestions and modified the manuscript accordingly. Please find more detailed answers to how we addressed your points below. Although a minor revision, we saw how the comments of the reviewers significantly improved the quality and intelligibility of the manuscript and, therefore again, want to express our gratitude.

Sincerely,

Patrick L. Jambura, MSc

Department of Palaeontology - Geozentrum

University of Vienna

Althanstrasse 14 - 1090 Vienna, AUSTRIA

E-mail: patrick.jambura@gmail.com

Reviewer #1: Dear Editor and Authors,

The study reports some interesting findings about the enigmatic genus Ptychodus from the Cretaceous of Spain and has a clear paleobiological approach. Despite the fossil material is fragmentary, it displays peculiar and useful characters. These characters have never been remarked in other papers and allow the authors to make some very interesting paleobiological inferences. Data are presented in an appropriate fashion and statistics are adequate, even though the Von Bertalanffy growth model was not applicable (but the authors clearly state the reason of its unsuitability). The inferences are well-supported and the language is fluent. All in all, this study is of international interest and provides novel and significant information.

There are only some minor points I would like to raise:

Reply: We want to let you know that we really appreciate your assessment. You raised a number of issues, which combined with your very helpful suggestions, significantly improved the quality of the manuscript. Please find more detailed responses on how we addressed your comments below.

- The title deals with first articulated specimens from Spain, but there is no mention in the manuscript to other findings of the genus Ptychodus from Spain. It would be useful to provide some other references to these findings and compare with the specimens reported herein.

Reply: Thank you for your suggestion. Due to comments of reviewer 2 we decided to remove this from the title. The manuscript now has its sole focus on the palaeecology of Ptychodus.

- The diagnostic vertebral characters reported here should be included in the diagnosis of the genus (see Hamm, 2020 and please emend).

Reply: thank you very much for this very helpful suggestion. A brief diagnosis paragraph on this genus’ vertebrae structure was added.

”Diagnosis (emended). For dental characters of this genus see Hamm [41]. Vertebral centra are well calcified, amphicoelous and circular, not dorso-ventrally compressed. Vertebrae display a calcification pattern of the asterospondylic type with four uncalcified areas radiating diagonally from the center to the bases of the neural and haemal arches. Numerous concentric lamellae extend outwards from the center. Underneath the smooth articular surfaces, parallel lamellae are oriented 360° around the center of the vertebrae and run perpendicular to the concentric lamellae”

- Please reconsider the structure of the manuscript and the issues with the images and other minor indications addressed in the annotated version of the manuscript attached.

Reply: We appreciate your suggestions and adopted them in our manuscript accordingly. Please find a detailed point-by-point response below.

- Line 10. alternatively consider "life history traits"

Reply: Reconsidered and changed accordingly.

- Line 71-73. The paper has clearly a paleobiological approach, but these characters has never been clearly evidenced in previous papers. This should be better remarked, maybe emending the diagnosis of the genus adding the characters evidenced in the analysis. For dental characters of the genus please refer to and complete the diagnosis by Hamm 2020. New Mexico Museum of Natural History and Science Bulletin 81.

Reply: We are very thankful for your suggestion and added a diagnosis paragraph accordingly.

- Line 78. I think that this subparagraph should be placed outside the Matherial and Methods as a single paragraph after the Introduction

Reply: We are aware that the Geological Setting often (but not exclusively) is part of the introduction. We, therefore, followed your suggestion and put it there. 

- Line 86. I would recommend the insertion of an image to better illustrate the stratigraphic framework of the section and the location of the specimen

Reply: Unfortunately, the section was measured by a diploma student back in the 1990s. The section is reproduced in his thesis but the labelling is in German. Moreover, we were not able to attain the original file for modification or permission by the author to modify the graph by ourselves. Unfortunately, the section on the coastline also differs slightly from other sections in the area (e.g., Gallemí et al. 2007), which we therefore can’t use. We consequently included the reference of the diploma thesis, which is available on request from the Free University of Berlin library service.

- Line 94. Please provide also more recent references. For example, Lamolda, M. A., Paul, C. R. C., Peryt, D., & Pons, J. M. (2014). The global boundary stratotype and section point (GSSP) for the base of the Santonian Stage," Cantera de Margas", Olazagutia, northern Spain. Episodes, 37(1), 2-13.

Reply: We followed your advice and added Gallemí J, López G, Martínez R, Pons JM. Macrofauna of the Cantera de Margas section, Olazagutia: Coniacian/Santonian boundary, Navarro-Cantabrian Basin, northern Spain. Cret Res. 2007;28: 5-17. doi: 10.1016/j.cretres.2006.05.014 and Lamolda et al. 2014.

- Line 169-171. It would be useful a brief introduction to the general structure of the shark centrum (corpus calcareum, intermedialia, etc.) (see e.g., Newbrey et al., 2015: p. 879)

Reply: We added a brief explanation of the centra composition that should help to better follow our description. We really appreciate your suggestion; the paragraph should be much clearer now.

“Shark centra form a double-cone calcification with densely calcified anterior and posterior conical ends, collectively referred to as corpora calcarea. Between the corpora calcarea is the intermedialia, which is softer than the corpus calcareum. The articular facet of the corpus calcareum is weathered and expose the inner layer of the vertebrae, showing concentric calcareous rings extending outwards from the center of the vertebrae. Numerous parallel lamellae are oriented perpendicular to these concentric lamellae (Fig 3A).”

- Line 169-171. Does the outer pattern of the corpus calcareum reflect the inner calcification pattern of the intermedialia? Apparently yes. Please take in consideration to better describe the external aspect for taxonomic comparison (Is it smooth or has some other pecularities? In lamniform sharks the external aspect is considered diagnostic for some species, e.g., the presence of papillose circular ridges on the surface of the corpus calcareum in Cardabiodon ricki)

Reply: We want to thank you for pointing this out. Actually, the described pattern cannot be seen on the surface of Ptychodus vertebrae but lay underneath a smooth articular surface (see for example Woodward 1911, Everhart & Caggiano 2004, Hamm 2019) and could only be seen because of the weathered condition of our material that exposed the inner layer. We modified our description to better reflect this, also the diagnosis paragraph that you suggested and was implemented, should further clarify this. 

“Anterior and posterior facets of the vertebral centra are weathered and expose the inner layer of the vertebrae, showing concentric calcareous rings extending outwards from the center of the vertebrae. Numerous parallel lamellae are oriented perpendicular to these concentric lamellae (Fig 3A).”

- Line 175. the area between the wedges, sensu Ridewood (1921)

Reply: Thank you very much for your suggestion. However, since this term has become a standard anatomical term in the last 100 years, it seemed redundant to us to explicitly state its origin in the manuscript and hasn’t been done in previous published work on shark vertebrae either. Therefore, we decided not to include the reference here (which is mentioned at a later point in the manuscript).

- Line 244. An image showing the band count would be useful to better illustrate the estimation.

Reply: Done. An image showing the band count has been included in the supplementary material.

- Line 274. an image illustrating upper and lower limit with a scale reference could potentially improve the visual effect of the paragraph (e.g., see Marramà et al., 2017: fig.5) doi.org/10.1080/08912963.2017.1341503

Reply: We appreciate your suggestion and agree with you on it better visualizing this paragraph. We, therefore, added such a figure (figure 4) to the paragraph.

- Line 329. PlosOne does not request a mandatory conclusion paragraph, but please consider making a paragraph remarking the most important points of this study. The title is "First articulated skeletal remains" but there is no reference to other findigs of Ptychodus in Spain and comparison with their kind of preservation (isolated teeth? isolated vertebral centra?). And the fact that these are the first articulated skeletal remains should be better remarked in the discussion or in the conclusions (there is a reference to this only in the title and in the introduction).

Reply: Thank you for your suggestion. As mentioned above, we followed the advice of reviewer 2 and modified the title and got rid of the “first articulated skeletal remains” part. This paper now solely focuses on the palaeoecological aspect of the find, which most important points are summarized in the last paragraph (even if it is not explicitly stated as “conclusion” paragraph, the “Inferred life history” paragraph basically represents the conclusions of this study).

- Fig 1. Please check the font and even out with the subsequent images

Reply: We very much appreciate your advice and standardized font and font size for all figures, using Times 10 and 18.

- Fig 2. I would recommend providing a scale bar at the bottom right corner of the four sections of the image

Reply: Done.

- Fig 3. I would recommend to center the scale reference in the middle of the scale bar, and I would suggest to make a thinner scale bar. Please check the font, even out with previous and next images

Reply: Done.

- Fig 4. Please even out the font of words and letters with that of previous images and check also previous images. PLOS indications: Arial, Times, or Symbol font only in 8-12 point.

Reply: Done.

Reviewer #2: The manuscript of Jambura & Kriwet is an interesting and important contribution to the knowledge of the paleobiology of the enigmatic extinct shark Ptychodus. The manuscript is of high-impact, well written, and provides sufficient data to suggest new hypotheses about the palaeobiology of Ptychodus. However, the manuscript still presents some issues that the authors should address before the publication. A further proofcheck is also needed to avoid small typos and errors. This is why I consider the manuscript worth of publication after minor revision. In particular:

Reply: We would like to express our gratitude for your assessment. We adapted most of your suggestions and hope to properly have addressed the remaining raising points. Please find a point-by-point response below.

- In the title and several times in the text (e.g. line 70) the authors emphasize that these are the first articulated Ptychodus remains. However, there are other specimens of Ptychodus described in the literature that can be considered as articulated, and these include nearly complete tooth sets (e.g. Amadori et al. 2019), and even articulated vertebrae (Hamm 2010, fig. 6B). This is why authors should remove ‘first’ from the title and subsequent statements in the manuscript.

Reply: We are sorry for causing confusion here: we are aware of other articulated remains of Ptychodus, this find however is the first report of such from Spain. To avoid any confusion, we removed “first” from the title. We kept it in the last paragraph of the introduction though, because here it should be clear that “first” refers to the locality, not the material (“Here we describe the first articulated shark remains from the Santonian of Spain, Europe.”)

- The authors used regression equations for extant galeomorph sharks to estimate the relationships between centrum diameter and total length of Ptychodus. However, the authors should clearly state, in Material and Methods, the reason why they use galeomorph sharks as comparative taxa, and not squalomorphs, or other extinct groups. As far as we know the affinities of Ptychodus are far from being clear. Is the presence of asterospondylic vertebrae enough to detect its close relationship with galeomorph sharks? Are there other hypotheses in the literature to be considered? If so why are these hypotheses discarded?

Reply: The choice of these three taxa was not based on their phylogenetic position, but on their large body size. We did not intend to suggest a close relationship between galeomorph sharks and Ptychodus (even if a similar mineralization pattern of the centra might indicate such a relation). However, squalomorph sharks generally are rather small species, with very few exceptions, like the six-gill shark Hexanchus griseus (TL 3-5m) and the sleeper sharks Somniosus spp. (TL 3-7m), but there are no regression models available for these species, making it impossible to use them as template species for this study. The reason why we choose not to pick regression models for smaller species can be found in our reply below. To prevent any inferences of the taxonomic position of Ptychodus, we chose to refer to our template species as “large extant sharks” instead of “large galeomorph sharks”. 

- Moreover, the authors use Carcharodon carcharias, Galeocerdo cuvier, Rhincodon typus as the only representatives of galeomorph sharks for comparisons. However, there are other extinct and living galeomorphs for which the relationship between centrum diameter and total body length are known, as reported by the authors themselves in the literature, that should be considered. For example: the extant Prionace glauca, Isurus oxyrhincus, Carcharhinus limbatus, and the extinct Carcharodon megalodon, and Cretoxyrhina mantelli (see Stevens 1975; Killam & Parsons 1989; Gottfried et al. 1996; Ribot-Carballal et al. 2005; Shimada 2008). There are probably even more taxa. In my opinion, the authors should consider for comparison the higher number of galeomorph taxa as possible and discuss, in any case, the results obtained. Otherwise, they should clearly state in Material and Methods the reason why they limited their comparisons to these three particular taxa.

Reply: Thank you for raising this point. The thought behind our decision to choose these three taxa was simple – we wanted to compare our specimen with extant species with similar vertebrae diameter. Following this approach we are expecting the most reliable size estimations for Ptychodus. Smaller sharks, like cat sharks that exhibit r-selected life history traits like fast growth can be expected to follow very different regression models, compared to large, K-selected sharks and therefore bias the estimated body size significantly. To test this hypothesis, we calculated the total length of our specimen based on the regression model for the blacktip sawtail catshark Galeus sauteri, which resulted in an estimated size of almost 9m (8,69m) – a value that is 20% higher than our suggested upper bound (and more than 200% of our lower bound). It is apparent that this is the result of different life history traits and lifestyles. Therefore, our decision to compare our specimen with sharks with similar sized vertebrae seems justified. Furthermore, these three template species are covering different groups, physiologies and ecologies, which resulted in a large interval of possible sizes and, therefore, already resembles a very careful and conservative interpretation. Fossil taxa like megalodon and Cretoxyrhina were not considered here, because TL for megalodon was calculated based on the ratio observed in white sharks and the TL of Cretoxyrhina was reconstructed (Shimada 2008 used “measurements” from Shimada 1997, which are partly reconstructed). We, therefore, followed your suggestion and added a brief note why we chose these three species in the M&M section.

“(2) We used published regression equations for large extant shark species with comparably sized vertebrae to estimate the relationship between centrum diameter (CD; mm) and total length (TL; cm). The following species with known regression equations were used as templates: (1) the great white shark Carcharodon carcharias [35]; (2) tiger shark Galeocerdo cuvier [37]; (3) whale shark Rhincodon typus [34].”

- Although it is not a common rule, several journals suggest to avoid the Saxon genitive in scientific papers being this mostly used in colloquial and informal sentences. I would suggest to avoid the Saxon genitive also here (see lines 27, 39, 57, 282, 312, 351).

Reply: Thank you very much for raising this point. We followed your suggestion and limited the use of the Saxon genitive to a minimum (1; line 27).

- Line 46. Replace “reproduction strategies” with “reproductive strategies”.

Reply: Done.

- Line 47. Some lamniform and carcharhiniform sharks and some rays (e.g. electric rays and stingrays) are actually ovoviviparous (aplacental viviparity). In my opinion this should be considered a different, third type of reproductive mode since it is quite different from the pure viviparity in that there is no placental connection and the unborn young are usually nourished by egg yolk.

Reply: We agree that the division into viviparity and oviparity is somehow superficial and does not cover the plethora of reproductive modes that can be found in elasmobranchs (extended oviparity, retained oviparity, aplacental viviparity, placental viviparity, etc.). However, neither would a subdivision into three categories, and a number of authors have actually abandoned the term ovoviviparity, as it comprises various modes of viviparity (e.g., Compagno 1990, Conrath & Musick 2012, Klimley 2013). Furthermore, a comparative data set of relative body size of the offspring of ovoviviparous and viviparous sharks is missing and would not allow any further interpretation of our data on this level. Therefore, we feel that a rough division into viviparity and oviparity best suits the interpretation of our data. Reconstructing if ptychodontid sharks were aplacental or placental viviparous is not possible at this point and would need an extensive review of reproductive modes and relative body sizes of offspring, which would represent a study in its own and is not within the scope of this manuscript.

- Line 50. Do you mean ‘slow growth’ instead of ‘small growth’?

Reply: Indeed, slow growth was meant here. Thank you very much for pointing this out.

- Line 104 to 107. This sentence is unclear and/or seems incomplete. Please re-write it.

Reply: Thanks a lot for highlighting this mistake. We changed this part accordingly and it now reads as follows: “Two disarticulated vertebral centra of varying preservational degree were collected and small sediment samples adjacent to the incomplete vertebral column were taken for screen-washing. The centra are housed in the fossil vertebrate collection of the Department of Palaeontology (University of Vienna) under the number EMRG-Chond-SK-1.”

- Line 134. I would write ‘conservative body shape’ instead of ‘consistent body form’, being ‘shape’ only related to the ‘outline’ but not to size.

Reply: Form was changed to shape. However, we are reluctant to call it “conservative”, since stem group elasmobranchs and early crown group elasmobranchs lived in shallow marine habitats and shared a similar body shape with today’s cat sharks. The pelagic realm was conquered later, therefore, the body shape associated with this lifestyle is rather derived than conservative.

- Line 141. It is unclear to me how the authors extrapolate the equation (1), or if this is based on a published paper. If the goal is using the following proportion:

TLEmrg-Chond-SK-1b : TLP.occidentalis = CDEmrg-Chond-SK-1b : CDP.occidentalis

you should consider that the product of the means (TLP.occidentalis x CDEmrg-Chond-SK-1b ) equals the product of the extremes (TLEmrg-Chond-SK-1b x CDP.occidentalis ) and then you must employ this equation:

TLemrg-Chond-SK-1b = (TLp.occidentalis x CDemrg-Chond-SK-1b) / CDp.occidentalis

Reply: The equation in question is a simple extrapolation that was established the same way you suggested it. The only difference is that we did not include parentheses, because the order of operations is irrelevant in this case, as multiplications and divisions follow the same priority rule and can be executed in any order. Therefore,

TLemrg-Chond-SK-1b = (TLp.occidentalis x CDemrg-Chond-SK-1b) / CDp.occidentalis 

gives the same results as our equation 

TLemrg-Chond-SK-1b = TLp.occidentalis x CDemrg-Chond-SK-1b/ CDp.occidentalis.

If TLp.occidentalis and CDemrg-Chond-SK-1b were added up or subtracted, this would be a whole other story of course.

Please also show somewhere which are the values from the published literature that you use for the equations. That means, just replace abbreviations (TL, CD) with numbers.

Reply: Thank you for your suggestion. The numbers were added to the extrapolation equation separately to better illustrate it. However, we had to use the variable X for TLp.occidentalis, because there are two values for the total length. To illustrate this, we added X€{150cm; 200cm}. The other equations were linear regression models, for which it is not applicable to replace abbreviations with numbers.

- Line 222-223. Should it maybe be ‘…and lack parallel lamellae ON vertebrae’ ?

Reply: Hamm (2010) did use “parallel lamellae vertebrae” as anatomical term in his work on Ptychodus rugosus to describe this structure. However, he never really defined this structure and we, therefore, decided to just describe this structure rather than naming it. 

- Line 255-256. ‘single vertebral centrum’ and ‘the largest vertebra’

Reply: Done.

- Line 258. In Shimada et al. (2009) it seems that the body size estimate for P. rugosus is not calculated based on vertebral centrum diameter, but rather on the antero-posterior tooth crown length. Please check.

Reply: Shimada et al. (2009) did use both, the vertebral centra diameter and the anterior-posterior length of the teeth to extrapolate the total length of P. rugosus. However, both approaches were based on the same, estimated TL of P. occidentalis and bear great risk to be erroneous. To clearly drive home our point here, we rewrote this paragraph to make these ambiguities clear to the reader.

“Based on the previously published centrum diameter and estimated total length of †Ptychodus occidentalis [16] we calculated an estimated total length of 887-1183cm for EMRG-Chond-SK-1. However, this estimation should be taken with caution, as the TL-CD relationship of †P. occidentalis is based on a single vertebral centrum which not necessarily represents the largest vertebra in this specimen and, therefore, can result in overestimated size approximations. Therefore, we recommend taking the previously estimated TL of 13m for †P. rugosus [16], which was also based on this TL-CD relationship, with much caution. Shimada et al. [16] also compared the anterior-posterior length of the teeth of †P. occidentalis and †P. rugosus and concluded that †P. rugosus might have reached a body size of even 14.4m. However, it is important to note here that the total length for †P. occidentalis, on which both calculations are based on, is unknown and was estimated based on the length of the lower jaw. Therefore, an erroneous size estimation for †P. occidentalis would also bias all subsequent calculations.”

- Line 290. Please spell the first time what CR is.

Reply: Done.

- In the whole manuscript, please be consistent in using meters or centimetres to indicate the total body length; and millimetres or centimetres for the radius/diameter of the centra.

Reply: We followed your advice and used cm for TL and mwm for CR/CD.

- Line 346-347. A bit unclear. What about ‘However, K-selected species are characterized by specific adaptations… etc’

Reply: Thank you very much for this suggestion. We adapted it and hope the sentence became clearer this way.

- Line 347/348. Replace ‘changing environments’ with ‘environmental changes’.

Reply: Done.

- Line 349. Be careful. You DID NOT demonstrate unambiguously that ptychodontids had K-selected traits (although you can assume/hypothesize it) because your hypothesis has been inferred based on indirect evidences and/or comparisons with living representatives, not with statistical demonstration or direct observations. Maybe better the sentence as 'We suggest/ can infer/ hypothesize that..."

Reply: We appreciate your suggestion and changed the sentence to tone it down a bit: “Our results strongly suggest that extinct ptychodontid sharks had K-selected traits, which in combination with a highly specialized trophic niche (durophagy) might have been a major intrinsic contributor to this group’s demise.”

- In figure 3, the names of the anatomical features should be in ‘lower case’.

Reply: Done.

- Line 498. The first author surname is ‘Larocca Conte’.

Reply: Thank you a lot for pointing this out. Corrected.

Suggested literature:

Hamm SA. 2010. The Late Cretaceous shark, Ptychodus rugosus , (Ptychodontidae) in the Western Interior Sea. Transactions of the Kansas Academy of Science, 113: 44-55.

Killam KA., Parsons, GR. 1989. Age and growth of the blacktip shark, Carcharhinus limbatus, near Tampa Bay, Florida. Fishery Bulletin. U.S. 87: 845-857.

Reply: Thank you very much for your suggestions. Hamm 2010 was already incorporated in the previous version of this manuscript. We agree that Killam and Parsons 1989 is a great study on the life history of Carcharhinus limbatus, and added it to our manuscript.

---

## [Editor Report · Decision Letter 1]

26 Mar 2020

Articulated remains of the extinct shark, Ptychodus (Elasmobranchii, Ptychodontidae) from the Upper Cretaceous of Spain provide insights into gigantism, growth rate and life history of ptychodontid sharks

PONE-D-20-05473R1

Dear Mr. Jambura,

We are pleased to inform you that your manuscript has been judged scientifically suitable for publication and will be formally accepted for publication once it complies with all outstanding technical requirements.

With kind regards,

Giorgio Carnevale, Ph.D

Academic Editor

PLOS ONE

---

## [Editor Report · Acceptance letter]

30 Mar 2020

PONE-D-20-05473R1 

Articulated remains of the extinct shark, Ptychodus (Elasmobranchii, Ptychodontidae) from the Upper Cretaceous of Spain provide insights into gigantism, growth rate and life history of ptychodontid sharks 

Dear Dr. Jambura:

I am pleased to inform you that your manuscript has been deemed suitable for publication in PLOS ONE. Congratulations! Your manuscript is now with our production department. 

With kind regards,

on behalf of

Dr. Giorgio Carnevale 

Academic Editor

PLOS ONE